# Deep learning-based correction of cataract-induced influence on macular pigment optical density measurement by autofluorescence spectroscopy

Akira Obana[1,2]*, Kibo Ote[3], Yuko Gohto[1], Hidenao Yamada[3], Fumio Hashimoto[3], Shigetoshi Okazaki[3], Ryo Asaoka[1]

1 Department of Ophthalmology, Seirei Hamamatsu General Hospital, Hamamatsu City, Shizuoka, Japan,
2 Department of Medical Spectroscopy, Institute for Medical Photonics Research, Preeminent Medical Photonics Education & Research Center, Hamamatsu University School of Medicine, Hamamatsu, Shizuoka, Japan, 3 Central Research Laboratory, Hamamatsu Photonics K.K., Hamamatsu, Shizuoka, Japan

* obana@sis.seirei.or.jp

## Abstract

### Purpose

Measurements of macular pigment optical density (MPOD) using the autofluorescence spectroscopy yield underestimations of actual values in eyes with cataracts. Previously, we proposed a correction method for this error using deep learning (DL); however, the correction performance was validated through internal cross-validation. This cross-sectional study aimed to validate this approach using an external validation dataset.

### Methods

MPODs at 0.25˚, 0.5˚, 1˚, and 2˚ eccentricities and macular pigment optical volume (MPOV) within 9˚ eccentricity were measured using SPECTRALIS (Heidelberg Engineering, Heidelberg, Germany) in 197 (training dataset inherited from our previous study) and 157 eyes (validating dataset) before and after cataract surgery. A DL model was trained to predict the corrected value from the pre-operative value using the training dataset, and we measured the discrepancy between the corrected value and the actual postoperative value. Subsequently, the prediction performance was validated using a validation dataset.

### Results

Using the validation dataset, the mean absolute values of errors for MPOD and MPOV corrected using DL ranged from 8.2 to 12.4%, which were lower than values with no correction ($P < 0.001$, linear mixed model with Tukey's test). The error depended on the autofluorescence image quality used to calculate MPOD. The mean errors in high and moderate quality images ranged from 6.0 to 11.4%, which were lower than those of poor quality images.

**Data Availability Statement:** All relevant data are within Supporting Information files.

**Competing interests:** A patent application on the deep learning-based correction technology was filed by Seirei Hamamatsu General Hospital and Hamamatsu Photonics K. K., a patent application number: WO2021/171788. There are no further patents, products in development, or marketed products to declare. The remaining authors declare no competing financial interests.

## Conclusion

The usefulness of the DL correction method was validated. Deep learning reduced the error for a relatively good autofluorescence image quality. Poor-quality images were not corrected.

## Introduction

The human retina contains a yellow pigment in the macula called macular pigment (MP). The MP consists of three xanthophyll-carotenoids: lutein [(3R, 3′R, 6′R)-lutein], zeaxanthin [(3R, 3′R)-zeaxanthin], and its stereoisomer *meso*-zeaxanthin [(3R, 3′S; *meso*)-zeaxanthin] [1,2]. Short-wavelength visible light (blue light) scatters not only in opacified cornea and crystalline lens with cataract but also within the retina. The macular pigment (MP) absorbs short-wavelength visible light and works to filter blue light. The absorption of blue light by MP improves contrast sensitivity and reduces glare disability [3–6]. Blue light also causes oxidative damage to photoreceptor cells and retinal pigment epithelial cells through photochemical reactions, and the oxidative damage is considered one of the important factors that cause age-related macular degeneration [4] [7–12]. Absorption of blue light and quenching oxygen radicals by MP are beneficial for suppressing the oxidative damage in the photoreceptor cells and retinal pigment epithelial cells [13–15]. Therefore, MPs are important for maintaining visual function and preventing eye diseases caused by photooxidative damage, such as age-related macular degeneration and accurate evaluation of MP is important for basic research and clinical studies. However, accurate measurements of MP in vivo are challenging, and there has not yet been clinical standardization for generalized practice.

There are several methods of measuring MP optical density (MPOD) [16]. Heterochromatic flicker photometry (HFP) is the most widely used subjective method; however, it requires training and cooperation of the patients, which is often difficult to obtain in older adults, and the examination time is relatively long [17–20]. Fundus reflectometry- [21] and fundus autofluorescence spectroscopy [22,23] are objective methods used in clinical studies. Fundus reflectometry is suitable for infants and children with clear ocular media [24,25] and fundus autofluorescence spectroscopy is suitable for adults because it uses the autofluorescence derived from lipofuscin, which accumulates in the retinal pigment epithelial cells with aging. The confocal scanning laser ophthalmoscope platform SPECTRALIS (Heidelberg Engineering, Heidelberg, Germany) is a commercially available device for two-wavelength autofluorescence spectroscopy. The details of the measurement principle were described by Delori et al. [22,23] Briefly, blue light (486 nm), which is strongly absorbed by MP, excites lipofuscin, and MPOD is derived by comparing the unattenuated intensities of lipofuscin fluorescence in the peripheral retina with the intensities attenuated by MP in the macular region, since MP is abundant in the macular region but sparse in the peripheral regions of the retina. The green light (518 nm) is weakly absorbed by the MP, and lipofuscin fluorescence by green light is used to compensate for background fluorescence. Using this device, MPOD levels at certain eccentricities and the spatial distribution and macular pigment optical volume (MPOV) [26] can be measured in a short time. The accuracy of the MPOD module of SPECTRALIS was validated by comparing it with HFP, and good concordance between SPECTRALIS and HFP was confirmed [17,27]. Good intra-examiner repeatability and inter-examiner reproducibility of the MPOD module of the SPECTRALIS were found [20,28]; however, the drawback of SPECTRALIS is that blue light excitation is vulnerable to transmission through ocular media. The attenuation of blue excitation light due to cataracts has been reported not to affect the measurement

of MPOD in HFP [29]. However, it does have an impact on the measurement by autofluorescence spectroscopy. Therefore, this drawback is evident when measuring MPOD in older adults. The color of the human crystalline lens changes from translucent to yellow and brownish with sclerosis of the lens nucleus and the lens cortex becomes cloudy owing to aging. These changes in color and opacity of the crystalline lens are the pathology of cataract. The blue excitation light is absorbed by the colored lens and scattered by the opacity of the lens cortex. Thus, the excitation of the SPECTRALIS is disturbed by absorption and scattering in the lens, and consequently, MP is underestimated in older adults. Additionally, in cataracts, the increased autofluorescence of the lens itself could potentially affect the measurements. In patients with dense cataracts, MPOD cannot be measured by SPECTRALIS because autofluorescence image quality is not good enough to calculate MPOD values. However, even in patients without dense cataracts whose autofluorescence images could be obtained, the MPOD was underestimated. In our previous studies [30,31], we measured MPOD at two time points: one with cataracts present and another after cataract surgery with intraocular lens implantation, comparing the values between the two conditions. We calculated the error as follows: error = (preoperative value–postoperative value)/ postoperative value. The mean absolute error of patients with mild cataracts (grade 0 and 1 of nuclear cataracts) [32] of those with best-corrected visual acuity of 1.0 (logMAR 0) or better was 18 ± 10%. These results suggested that MP measurement is affected by lens aging in many older adults. Therefore, we proposed three correction methods to compensate for this underestimation. The first is an objective method using a regression equation with age, nuclear cataract grade, and imaging quality index as independent variables [30]. The second is a subjective method that classifies autofluorescence images into three grades and adopts correction factors (CFs) for each grade [30]. The third is a correction method using deep learning (DL) [31]. In a previous study [31], we evaluated the accuracy of these three methods in 197 eyes. The results showed that the errors were 29% without any correction, 14% using the regression equation method, 10% using the subjective classification method, and 8.7% using the DL method when we predicted MP volumes in the central area of the retina. The DL method was statistically superior to the regression equation method and similar to the subjective classification method. Since the correction method using DL is easy to perform, we proposed the application of this method for the measurement of MP in older adults. However, in our previous study [31], the correction performance was validated through internal cross-validation, and further validation was required using an external validation dataset. In the present study, we validated the correction performance of the DL approach using a new external validation dataset.

## Methods

### Patients and measurement of MPOD

In this cross-sectional study, we enrolled patients who underwent cataract surgery with yellow-tinted intraocular lens implantation at the Seirei Hamamatsu General Hospital between September 2019 and January 2022. In a previous study [31], we evaluated the accuracy of DL in 197 eyes of 148 patients using leave-one-case-out cross-validation. In the present study, this dataset was used as a training dataset. In addition, we conducted the same measurements in newly recruited 157 eyes of 105 patients, which were used as the validation dataset. The demographics of the validation dataset are listed in Table 1 (For this information in the training dataset, please refer to our previous paper [31]).

All the patients underwent visual acuity testing and intraocular pressure measurements. Slit-lamp and fundus examinations, fundus photography, optical coherence tomography (OCT) (SPECTRALIS OCT, Heidelberg Engineering), and measurement of MPOD levels

Table 1. Demographics of the validation dataset.

| Variables | Values (n = 157) |
|---|---|
| Subject number | 105 (M48, F57) |
| Age range (years) | 42–90 |
| Mean age (SD) (years) | 73.0 (9.0) |
| Number of eyes of each sex | M 71, F 86 |
| Mean logMAR before surgery (SD) | 0.09 (0.14) |
| Number of eyes of subjective classification of autofluorescence image (%) | |
| High quality | 22 (14) |
| Moderate quality | 41 (26) |
| Poor quality | 94 (60) |

SD, standard deviation.

were performed within 2 weeks before surgery and 4 or 5 days after surgery. The measurement of MPOD was performed under mydriasis induced by 2.5% phenylephrine hydrochloride and 1% tropicamide. The average optical densities at 0.25˚, 0.5˚, 1˚, and 2˚ eccentricities (local MPODs) and the MPOV in the central retina within 9˚ eccentricity were analyzed as previously reported [30,31,33,34]. The cutoff eccentricity was set at 9˚. Autofluorescence images were subjectively classified into three categories depending on the image quality: relatively high, moderate, and poor, as described in the previous study [30].

This case series was approved by the Institutional Review Board of the Seirei Hamamatsu General Hospital (IRB No. 2251,). The study was conducted in accordance with the principles of the Declaration of Helsinki. All patients provided written informed consent before enrolment.

## Finding predicted CFs using deep learning

The details are described in our previous paper [31]; however, we input autofluorescence images of SPECTRAIS OCT by blue and green light and subtraction images of these two into pre-trained VGG16 network [35] and fine-tune the network to predict CFs for local MPOD at four eccentricities and MPOV. This model was trained using a training dataset, and the correction performance was investigated using a validation dataset.

## Statistical analyses

Differences in all local MPODs and MPOV before and after surgery were compared using a linear mixed model. The details of the linear mixed model are described in our previous study [31], in which the linear mixed model was adjusted for the hierarchical structure of the data (each eye was nested to each patient), modeling in a way in which measurements were grouped within patients to reduce the possible bias derived from the nested structure of the data. The differences in the mean absolute values of the error of MPOD and MPOV corrected using DL were compared among the image qualities using the linear mixed model. Statistical significance was set at $P < 0.05$.

## The definition of the terms

In our previous papers [30,31], the MPOD values after surgery were labeled as the "true value". However, the measurements obtained with the current instrument do not directly assess pigment density within the tissue, and thus, do not represent the true value in the strictest

meaning. Therefore, in this paper, we designated the postoperative MPOD(MPOV) values as MPOD(MPOV)$_{after}$, the preoperative values as MPOD(MPOV)$_{before}$, and the corrected values as MPOD$_(MPOV)_{corrected}$.

## Results

Cataract surgery was performed without any complications, and intraocular lenses were fixed to the lens capsule in all eyes. No eye had severe intraocular inflammation, corneal disorders, or increased intraocular pressure at the time of measurement of MPOD levels after surgery.

The local MPOD at the four selected eccentricities and MPOV before and after surgery in the validation dataset are shown in Table 2. Local MPOD$_{before}$ and MPOV$_{beofore}$ were lower than those after surgery in all eyes, with some exceptions. The number of eyes that were over-estimated before surgery was 2, 4, 2, 3, and 9 for local MPOD at 0.25˚, 0.5˚, 1˚, 2˚, and MPOV, respectively. The mean local MPODs and MPOV were significantly lower before than after surgery ($P < 0.001$, linear mixed model).

### CFs and errors

The predicted CFs for the local MPODs at the four eccentricities and MPOV are listed in Table 3. The actual errors of MPOD(MPOV) were calculated as actual error = {MPOD (MPOV)$_{before}$−MPOD(MPOV)$_{after}$}/MPOD(MPOV)$_{after}$. Errors in MPOD(MPOV) corrected using DL were calculated in the same manner: error = {MPOD(MPOV)$_{corrected}$−MPOD (MPOV)$_{after}$} / MPOD(MPOV)$_{after}$. Table 4 shows the absolute values of actual errors and errors. Table 5 shows the absolute value of the error of MPOD(V)$_{corrected}$ using DL depending on the subjective classification of the quality of autofluorescence images (Refer to Fig 1 in our previous paper [30]). Error was significantly lower in high and moderate image qualities for all eccentricities (except for 1˚ and MPOV) than poor quality images.

## Discussion

Local MPODs and MPOV were underestimated in all eyes, with some exceptions. The mean local MPOD at 0.25˚, 0.5˚, 1˚, and 2˚ eccentricities and MPOV before surgery were

**Table 2. Local MPOD and MPOV before and after surgery.**

| | Before surgery<br>Min-Max<br>Median<br>Mean ± SD | After surgery<br>Min-Max<br>Median<br>Mean ± SD | P-value |
|---|---|---|---|
| **MPOD at 0.25˚** | 0.13–0.98<br>0.49<br>0.49 ± 0.15 | 0.24–1.33<br>0.82<br>0.80± 0.19 | < 0.001 |
| **MPOD at 0.5˚** | 0.12–0.90[‡]<br>0.47<br>0.48 ± 0.14 | 0.32–1.21[‡]<br>0.74<br>0.74 ± 0.18 | < 0.001 |
| **MPOD at 1˚** | 0.08–1.11[‡]<br>0.47<br>0.47 ± 0.13 | 0.27–1.17<br>0.71<br>0.70 ± 0.15 | < 0.001 |
| **MPOD at 2˚** | 0.06–0.52[‡]<br>0.25<br>0.26 ± 0.08 | 0.09–0.74<br>0.34<br>0.34 ± 0.11 | < 0.001 |
| **MPOV** | 3,451–32,041<br>15,732<br>15,618 ± 4,915 | 6,082–40,224[‡]<br>19,646<br>19,754 ± 5,792 | < 0.001 |

MPOD: Macular pigment optical density; MPOV: Macular pigment optical volume; SD: Standard deviation, [‡]Values showed normal distribution.

**Table 3. Predicted CFs for local MPOD and MPOV.**

| | Predicted CF Min-Max Median Mean ± SD |
|---|---|
| **MPOD at 0.25˚** | 1.16–2.67 1.58 1.67 ± 0.34 |
| **MPOD at 0.5˚** | 1.12–2.49 1.49 1.57 ± 0.30 |
| **MPOD at 1˚** | 1.11–2.30 1.43 1.49 ± 0.26 |
| **MPOD at 2˚** | 1.05–1.77 1.29 1.32 ± 0.16 |
| **MPOV** | 1.02–1.63 1.23 1.26 ± 0.13 |

MPOD: Macular pigment optical density; MPOV: Macular pigment optical volume; CF: Correction factor; DL: Deep learning; SD: Standard deviation.

significantly lower than those after surgery in the present 157 eyes, similar to those in the previous 197 eyes. The mean actual error without correction ranged from 21.2% to 39.0% for local MPODs and 20.7% for MPOV. In contrast, correction using DL achieved a mean error of 9.4% to 12.4% for local MPOD and 8.2% for MPOV. The mean ± SD of actual error in the present validation dataset were 12.4 ± 10.9 [12.7 ± 10.4], 11.5 ± 10.9 [11.4 ± 8.6], 11.7 ± 14.6 [10.8 ± 8.8], 9.4 ± 8.5 [8.0 ± 7.2], and 8.2 ± 8.2 [7.9 ± 8.2] %, and the median were 9.5 [10.1],

**Table 4. Absolute values of error of MPOD and MPOV with no correction and with correction using DL.**

| | Error (%) with no correction Min-Max Median Mean ± SD | Error (%) corrected using DL Min-Max Median Mean ± SD |
|---|---|---|
| **MPOD at 0.25˚** | 0–75.0 37.8 39.0 ± 15.7 | 0.10–46.6 9.5 12.4± 10.9 |
| **MPOD at 0.5˚** | 1.96–72.9 33.8 34.2 ± 15.3 | 0.03–77.0 8.9 11.5 ± 10.9 |
| **MPOD at 1˚** | 0–104.3 31.8 32.9 ± 15.7 | 0.06–147.7 8.0 11.7 ± 14.6 |
| **MPOD at 2˚** | 0–66.7 21.7 24.0 ± 12.8 | 0.04–52.5 6.6 9.3 ± 8.5 |
| **MPOV** | 0.27–76.7 18.8 21.2 ± 11.3 | 0–69.0 5.8 8.2 ± 8.2 |

MPOD, macular pigment optical density; MPOV, macular pigment optical volume; DL, deep learning; SD, standard deviation.

**Table 5. Absolute values of error of MPOD and MPOV corrected using modified DL depending on the quality of autofluorescence image.**

| Image quality | High and moderate qualities<br>Min-Max<br>Median<br>Mean ± SD | Poor quality<br>Min-Max<br>Median<br>Mean ± SD | *P*-value |
|---|---|---|---|
| **MPOD at 0.25˚** | 0.4–41.4<br>7.2<br>8.9 ± 7.9 | 0.1–46.6<br>12.8<br>14.7 ± 12.0 | 0.0020 |
| **MPOD at 0.5˚** | 0.15–31.5<br>6.9<br>8.5 ± 6.6 | 0.03–77.0<br>10.7<br>13.5 ± 12.7 | 0.0090 |
| **MPOD at 1˚** | 0.08–147.7<br>7.3<br>11.4 ± 20.1 | 0.06–48.3<br>9.4<br>11.8 ± 9.3 | 0.50 |
| **MPOD at 2˚** | 0.09–40.4<br>5.5<br>7.6 ± 7.5 | 0.04–52.5<br>7.8<br>10.5 ± 9.0 | 0.033 |
| **MPOV** | 0.04–25.6<br>4.7<br>6.0 ± 5.2 | 0.003–69.0<br>6.8<br>9.7 ± 9.5 | 0.0054 |

*P*-value; Linear mixed model.

MPOD, macular pigment optical density; MPOV, macular pigment optical volume; DL, deep learning; SD, standard deviation.

8.9 [10.1], 8.0 [9.0], 6.6 [6.4], and 5.8 [6.0] % for MPOD at 0.25˚, 0.5˚, 1˚, 2˚ and MPOV, respectively. The results of the present validation dataset are similar to those of a previous training dataset [31]. In case of high and moderate qualities of autofluorescence images, the mean errors of the present DL method ranged from 7.6% to 11.4% for local MPOD and 6.0% for MPOV, and these were significantly lower than the mean errors in case of poor quality. The effectiveness of the correction method using DL was confirmed using an external validation dataset, and the correction was efficient, especially for cases with relatively good image quality, equivalent to high and moderate qualities in our classification [30].

Correction factors increased toward the center of the fovea, as shown in previous studies [27,30,31]. The actual errors were larger for the local MPOD closer to the center. This is because the crystalline lens was thicker at the center, and the blue excitation light was highly absorbed at the center of the lens. The errors in MPOD corrected by DL were larger in the local MPOD closer to the foveal center. Thus, compensation is difficult for local MPOD close to the foveal center, even with correction by DL. In contrast, the error of MPOV was smaller than that of the local MPOD. The local MPOD at certain eccentricities has been investigated in many studies; however, the distribution of the MP varies among patients [36]. Therefore, MPOV has been proposed as a suitable value for investigating MP [26]. The present correction is suitable for this proposal, as MPOV is less affected by lens opacity and more accurately corrected using the DL method.

The errors in MPOD and MPOV in the correction using DL depended on the quality of the autofluorescence image. Examples of high-, moderate-, and poor-quality images are shown in our previous report (Refer to Fig 1 in our previous report [30]). In the case of subjectively classified high- and moderate-quality images, the mean errors of MPOD and MPOV were significantly smaller than those of poor-quality images. This suggests that correction with DL cannot compensate for the deficiency of the original image. The mean absolute error of MPOV was 6.0 ± 5.2%, and the median was 4.7% in the case of high and moderate image qualities. The

acceptable error for clinical use was uncertain; the smaller the better; however, we considered that this value was easier to accept compared with 9.7% (median, 6.8%) for poor-quality images. Therefore, we propose a hybrid correction using the subjective classification of auto-fluorescence images and the DL method to estimate MPOV. Of the present 157 eyes, high- or moderate-quality images were obtained in 63 eyes (40.1%). Out of the 354 eyes, including the 197 eyes in the training dataset, high- or moderate-quality images were obtained in 201 eyes (56.8%). These values suggest that the DL method can be applied to about half of cases which have relatively mild cataract.

This study had several limitations. First, even after DL correction, the error was not negligible in poor-quality autofluorescence images, and the present correction method using DL was applicable to relatively good-quality images. Second, we adopted $MPOD_{after}$ measured shortly after surgery because our previous study [37] suggested that MPOD changes depend on the postoperative elapsed time. However, there might be some influence on the measurement of MPOD due to surgical invasion, although all surgeries were performed without any complications, and no severe postoperative inflammation was noted. In general, the accuracy of DL and the proposed algorithm can be improved with larger training data. We only used preoperative images to train the convolutional neural network (CNN). If we can use postoperative images to train the CNN through self-supervised learning, we can double the dataset size. This is an important topic for future studies. Finally, the present results are based on data from Japanese individuals at a single facility, and to generalize these results, a multi-facility study including individuals of different races is necessary.

To obtain MPOD and MPOV with less bias in older adults, a compensation method is required. The correction method using DL is easy to perform and can reduce errors. This study validates the accuracy of the DL method. However, even with the present correction method, the error was relatively large in MPOD at eccentricities close to the foveal center and in cases of poor autofluorescence image quality. Therefore, the present DL correction was suitable for estimating the MPOV in cases of relatively good image quality.

## Supporting information

**S1 Data set. We have added STROBE-statement and minimal data set to supplemental information.**
(XLSX)

**S1 File. STROBE statement—Checklist of items that should be included in reports of *cross-sectional studies*.**
(PDF)

## Acknowledgments

The authors would like to thank Etsuji Yoshikawa, Department of Medical Spectroscopy, Institute for Medical Photonics Research, Preeminent Medical Photonics Education & Research Center, Hamamatsu University School of Medicine for his useful comments on this study.

## Author Contributions

**Conceptualization:** Akira Obana, Kibo Ote, Hidenao Yamada, Shigetoshi Okazaki, Ryo Asaoka.

**Data curation:** Akira Obana.

**Formal analysis:** Kibo Ote, Fumio Hashimoto, Ryo Asaoka.

**Investigation:** Akira Obana, Yuko Gohto.

**Supervision:** Ryo Asaoka.

**Writing – original draft:** Akira Obana, Kibo Ote.

**Writing – review & editing:** Hidenao Yamada, Ryo Asaoka.

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
