## [Decision Letter · Decision Letter 0]

3 Nov 2023

PONE-D-23-31420Deep Learning-based Correction of Cataract-induced Influence on Macular Pigment Optical Density Measurement by Autofluorescence TechniquePLOS ONE

Dear Dr. Obana,

Thank you for submitting your manuscript to PLOS ONE. After careful consideration, we feel that it has merit but does not fully meet PLOS ONE’s publication criteria as it currently stands. Therefore, we invite you to submit a revised version of the manuscript that addresses the points raised during the review process.

We look forward to receiving your revised manuscript.

Kind regards,

Tatsuya Inoue

Academic Editor

PLOS ONE

Journal Requirements:

4. Thank you for stating the following financial disclosure: "This research was supported in part by Hamamatsu Photonics K.K." 

6. We note that you have indicated that data from this study are available upon request. PLOS only allows data to be available upon request if there are legal or ethical restrictions on sharing data publicly. For more information on unacceptable data access restrictions, please see http://journals.plos.org/plosone/s/data-availability#loc-unacceptable-data-access-restrictions. 

8. We are unable to open your Supporting Information file "STROBE-checklist-v4-cross-sectional.doc". Please kindly revise as necessary and re-upload.

Reviewers' comments:

Reviewer's Responses to Questions

**Comments to the Author**

1. Is the manuscript technically sound, and do the data support the conclusions?

Reviewer #1: Yes

Reviewer #2: No

2. Has the statistical analysis been performed appropriately and rigorously? 

Reviewer #1: Yes

Reviewer #2: Yes

3. Have the authors made all data underlying the findings in their manuscript fully available?

Reviewer #1: Yes

Reviewer #2: Yes

4. Is the manuscript presented in an intelligible fashion and written in standard English?

Reviewer #1: Yes

Reviewer #2: Yes

5. Review Comments to the Author

Reviewer #1: The authors reported the efficacy of deep learning-based correction of cataract-induced errors of macular pigment optical density (MPOD) calculated by the subtraction of autofluorescence on a fluorescein angiography device. As the authors stated, MPOD measurement is important to investigate impacts of MPOD on development and progression of age-related macular degeneration. Cataract interferes with capture of high quality images of fundus autofluorescence. They insisted on usefulness of deep learning to correct measured values. The manuscript was succinctly written. However, there may be some issues to be addressed. First of all, it may be necessary to explain the impacts of cataract and intraocular lens on autofluorescence imaging so that general readers can understand the content easily. Although the authors stated that cataract interferes excitation light transmission, it may be also critical that cataract itself has autofluorescence, increasing background density of captured images. The authors should discuss this issue. Also, there are two types of intraocular lens (IOL): clear and yellow-tinted. The latter may block somewhat blue light transmission, potentially affecting captured images of autofluorescence. Did all patients undergo the implantation of the same type IOL? It would be better to provide representative images before and after cataract surgery. Second, it would be better to provide graphs to show the relationship of age and MPOD before and after surgery and that of cataract status and MPOD.

(Lines 2, 56) The term, ‘autofluorescence technique’ should be ‘the autofluorescence subtraction method’.

(line 66) The phrase, ‘local MPODs at 0.25…’ should be attached to the first sentence, because this is the method of measurement.

(line 73) The word, ‘qualities’ should be amended to ‘quality images’.

(line 94) The reference (4) should be placed correctly.

(line 95) For general readers, the authors should describe the details of basic research and clinical studies (, exemplified like ‘to investigate potential for disease prophylaxis and assess pigment density in individuals’).

(line 103) The term, ‘fluorescence of lipofuscin’ should be ‘autofluorescence derived from lipofuscin’. Also, ‘ the retina’ should be ‘retinal pigment epithelial cells’.

(line 121) The authors stated that the blue excitation light is absorbed by the colored lens. How is the influence of yellow-tinted IOL? Also, the autofluorescence of the colored lens may interfere with captured images of fundus autofluorescence. The authors should discuss these issues.

(line 148) It would be better if the authors could discuss not only the effectiveness of further validation but also the relationship of macular pigments to age and cataract status.

(line 211) Is ‘(true value- nominal value) / true value’ right?

(lines 216 and 231; and Table 5) The numbers of high or moderate quality images and poor quality image should be shown. Was there difference in cataract status between these groups? The authors should discuss possible causes of poor qualities. It would be better if the authors could provide representative images.

(line 220) It would be better if the authors could explain the details of some exceptions.

Reviewer #2: This paper is interesting but has a number of challenges that I have outlined below.

The objective of the study is to determine whether an algorithmic correction can improve MP measures made using the AF technique. The issue, according to the authors, is that AF is confounded by cataracts and alters MP values as a result. The problem is that a dense anterior lens absorbs (and scatters) the measuring light (460 nm) and even some of the reference leading to an overall reduction in MP values. The “smart” correction the authors apply is one that, they argue, can most accurately adjust the MP value back to its true value.

1. One problem in the study (and in previous studies) is that in order to validate this correction, the authors would need to know what the “true” value of MP actually is (not “defined” as true, but actually true). If you are trying to validate a method, it needs to be compared against a known standard. The best way to do this would have been to measure the lens and MP density in adults with healthy eyes (even younger people have large variation in lens density) using validated methods (Like HFP for MP) and then see how the method/corrections works in this best scenario.

2. Why is deep learning (or other corrections) better than simply measuring the density and scattering of the lens at the wavelengths used to measure MP? Even if that is too laborious, wouldn’t you want to do this at least for the purpose of checking the accuracy of your derivations?

3. How do you know it is only the lens confounding AF measures? AF relies on a measuring substrate (lipofuscin) that changes with age and often declines sharply with progressing retinal disease.

4. The original AF method required that the retina is bleached. Was this done? Doesn’t a cataractous lens effect the completeness of the bleach? How do subjects fixate when making peripheral measures if their retina is bleached and they cannot see?

5. The MPOD values in Table 2 seem unrealistic. First, the spatial distribution of values seems very flat (MP is much more leptokurtic). Second, the AF measurement is not at peak MPOD, it is at 486 nm. Your measure at 0.5 degrees (analogous to the standard 1 degree diameter), after surgery, was 0.74. Compare this to Ciulla et al. (2001, IOVS) who also measured MPOD before and after cataract (it is good scholarship to actually cite studies that do the same work). In their group of patients, MPOD was 0.206 before and 0.18 after cataract extraction.

6. PLOS authors have the option to publish the peer review history of their article (what does this mean?). If published, this will include your full peer review and any attached files.

Reviewer #1: **Yes: **Tsutomu Yasukawa

Reviewer #2: No

---

## [Author Response · Author response to Decision Letter 0]

24 Nov 2023

The response to the reviewers has been documented in a separate file. Please refer to that for details.

---

## [Decision Letter · Decision Letter 1]

12 Jan 2024

PONE-D-23-31420R1Deep Learning-based Correction of Cataract-induced Influence on Macular Pigment Optical Density Measurement by Autofluorescence SpectroscopyPLOS ONE

Dear Dr. Obana,

Thank you for submitting your manuscript to PLOS ONE. After careful consideration, we feel that it has merit but does not fully meet PLOS ONE’s publication criteria as it currently stands. Therefore, we invite you to submit a revised version of the manuscript that addresses the points raised during the review process.

We look forward to receiving your revised manuscript.

Kind regards,

Tatsuya Inoue

Academic Editor

PLOS ONE

Journal Requirements:

Reviewers' comments:

Reviewer's Responses to Questions

**Comments to the Author**

1. If the authors have adequately addressed your comments raised in a previous round of review and you feel that this manuscript is now acceptable for publication, you may indicate that here to bypass the “Comments to the Author” section, enter your conflict of interest statement in the “Confidential to Editor” section, and submit your "Accept" recommendation.

Reviewer #1: All comments have been addressed

Reviewer #2: (No Response)

Reviewer #3: All comments have been addressed

2. Is the manuscript technically sound, and do the data support the conclusions?

Reviewer #1: Yes

Reviewer #2: No

Reviewer #3: Yes

3. Has the statistical analysis been performed appropriately and rigorously? 

Reviewer #1: Yes

Reviewer #2: N/A

Reviewer #3: Yes

4. Have the authors made all data underlying the findings in their manuscript fully available?

Reviewer #1: Yes

Reviewer #2: Yes

Reviewer #3: Yes

5. Is the manuscript presented in an intelligible fashion and written in standard English?

Reviewer #1: Yes

Reviewer #2: Yes

Reviewer #3: Yes

6. Review Comments to the Author

Reviewer #1: (No Response)

Reviewer #2: Unfortunately, there are simply too many errors in the revisions and in the responses by the authors for me to recommend publication. Here are a couple of examples.

1. In the revisions, the authors write "...The macular pigment (MP) absorbs short-wavelength

93 visible light and works to filter blue light. Blue light is highly scattered in the retina... The MP absorbs short-wavelength light AND works to filter blue light? Those are basically the same thing. Also, blue light is mostly scattered in the anterior media (the lens) not the retina. These kind of small technical errors are pretty pervasive.

2. In your response letter, you write "...Clinically, as far as we know, there is no method to measure light transmittance and scattering intensity of the crystalline lens." There are many fairly easy methods for measuring the density and scattering of the lens. Further, I am not sure why the method would need to be based on a clinical instrument. This is a research study. You can use more elaborate methods first and then if the method is valid it can be translated for use within the clinic.

Reviewer #3: The authors used DL to correctly evaluate the MPOD values of cataractous eyes measured by SPECTRALIS and again validated its usefulness with an external validation data set. Although only high-quality images are applicable, the usefulness of DL is well demonstrated. If possible, what about adding a graph (e.g. box-and-beard diagram) to facilitate visual understanding of the data?

7. PLOS authors have the option to publish the peer review history of their article (what does this mean?). If published, this will include your full peer review and any attached files.

Reviewer #1: **Yes: **Tsutomu Yasukawa

Reviewer #2: No

Reviewer #3: No

---

## [Author Response · Author response to Decision Letter 1]

16 Jan 2024

Reply to reviewers.

Dear Reviewer #2

Thank you so much for conducting the second review. However, we regret that our responses to the first review did not seem to obtain your full understanding. What we are seeking is a relatively simple method to measure MP which can be used in outpatient ophthalmic care. Spectralis is very widely used as an OCT examination, and measuring MP with Spectralis is very convenient for ophthalmologists. However, as discussed in the previous reports and this paper, Spectralis has the drawback that excitation light is affected by cataracts.

 In our previous report, we reported that using AI to assess the image quality of autofluorescence photographs for calculating MP can reduce measurement errors. However, since the previous report only validated the method among internal data, in this study, we validated this approach using external data. The main point of this paper lies in proving the validity of the correction method using AI through external validation. The purpose of this paper is not to study the value of MP, factors involved in MP, or the function of MP. These aspects have been reported by many researchers, including in our previous work.

 Taking the above into consideration, I will respond to your criticisms.

1. In the revisions, the authors write "...The macular pigment (MP) absorbs short-wavelength 93 visible light and works to filter blue light. Blue light is highly scattered in the retina... The MP absorbs short-wavelength light AND works to filter blue light? Those are basically the same thing. Also, blue light is mostly scattered in the anterior media (the lens) not the retina. These kind of small technical errors are pretty pervasive.

Thank you for your comment, but we respectfully disagree with you.

UV-B and C do not transmit to the retina, but blue visible light definitely transmits to the retina and is captured by the blue-sensitive cones. Blue light is more prone to scattering within tissues compared to longer-wavelength light, and MP suppresses scattered light within the retina. It has been reported that individuals with higher amount of MP exhibit higher contrast sensitivity (Nolan JM. IOVS 2016;57:3429), and the reason for this is supposed to be the suppression of scattered light by MP. However, as you pointed out, blue visible light scatters in the corneal opacity and cataract, so we have revised the description in line 92-93. We apologize for any misunderstanding caused by our initial description.

2. In your response letter, you write "...Clinically, as far as we know, there is no method to measure light transmittance and scattering intensity of the crystalline lens." There are many fairly easy methods for measuring the density and scattering of the lens. Further, I am not sure why the method would need to be based on a clinical instrument. This is a research study. You can use more elaborate methods first and then if the method is valid it can be translated for use within the clinic.

In the clinical setting of Ophthalmology, the severity of cataracts is determined by ophthalmologist through the observation of the lens using a slit lamp, following international grading system (Thylefors B. Ophthalmic Epidemiol 2002;9:83). This subjective assessment method is also utilized in many clinical studies. Although our previous report utilized a Scheimpflug camera, it is still a subjective judgement method. If there were a device that could be applied to human and safely measure the light transmission of the crystallin lens, it would be possible to correct the measurement errors of MP based on light transmission. This would be an ideal correction method, as you suggested. However, to the best of our knowledge, such a device has not been commercially available, and hence we have to seek for a method to measure MP that can be readily used at clinical practice rather than in basic research. Therefore, we explored a method of using AI to assess the image quality of autofluorescence photographs. If you are aware of a method for measuring the light transmission of the crystallin lens that can be used in clinical practice, we would appreciate it if you could share the information with us. We want to apply it in our next study.

Dear Reviewer #3

We appreciate your review work and thanks for your positive comments.

Regarding the instruction to add a boxplot, in the previous report, boxplots related MP before and after cataract surgery were included. Although it is possible to create similar boxplots with the data from this paper, we have chosen to omit them as they would be highly redundant. We appreciate your understanding.

---

## [Decision Letter · Decision Letter 2]

21 Jan 2024

Deep Learning-based Correction of Cataract-induced Influence on Macular Pigment Optical Density Measurement by Autofluorescence Spectroscopy

PONE-D-23-31420R2

Dear Dr. Obana,

We’re pleased to inform you that your manuscript has been judged scientifically suitable for publication and will be formally accepted for publication once it meets all outstanding technical requirements.

Kind regards,

Tatsuya Inoue

Academic Editor

PLOS ONE

Additional Editor Comments (optional):

Reviewers' comments:

Reviewer's Responses to Questions

**Comments to the Author**

1. If the authors have adequately addressed your comments raised in a previous round of review and you feel that this manuscript is now acceptable for publication, you may indicate that here to bypass the “Comments to the Author” section, enter your conflict of interest statement in the “Confidential to Editor” section, and submit your "Accept" recommendation.

Reviewer #3: (No Response)

2. Is the manuscript technically sound, and do the data support the conclusions?

Reviewer #3: (No Response)

3. Has the statistical analysis been performed appropriately and rigorously? 

Reviewer #3: (No Response)

4. Have the authors made all data underlying the findings in their manuscript fully available?

Reviewer #3: (No Response)

5. Is the manuscript presented in an intelligible fashion and written in standard English?

Reviewer #3: (No Response)

6. Review Comments to the Author

Reviewer #3: (No Response)

7. PLOS authors have the option to publish the peer review history of their article (what does this mean?). If published, this will include your full peer review and any attached files.

Reviewer #3: No

---

## [Editor Report · Acceptance letter]

5 Feb 2024

PONE-D-23-31420R2 

PLOS ONE

Dear Dr. Obana, 

I'm pleased to inform you that your manuscript has been deemed suitable for publication in PLOS ONE. Congratulations! Your manuscript is now being handed over to our production team.

Kind regards, 

on behalf of

Dr. Tatsuya Inoue 

Academic Editor

PLOS ONE